# Self-Sensing Variable Stiffness Actuation of Shape Memory Coil by an Inferential Soft Sensor

**DOI:** 10.3390/s23052442

**Published:** 2023-02-22

**Authors:** Bhagoji Bapurao Sul, Dhanalakshmi Kaliaperumal, Seung-Bok Choi

**Affiliations:** 1Department of Instrumentation and Control Engineering, National Institute of Technology, Tiruchirappalli 620015, Tamil Nadu, India; 2Department of Mechanical Engineering, The State University of New York, Korea (SUNY Korea), Incheon 21985, Republic of Korea; 3Department of Mechanical Engineering, Industrial University of Ho Chi Minh City (IUH), Ho Chi Minh City 70000, Vietnam

**Keywords:** shape memory coil, joule heating effect, self-sensing actuation, variable stiffness actuation, electrical resistance, support vector machine regression model, nonlinear regression model

## Abstract

Self-sensing actuation of shape memory alloy (SMA) means to sense both mechanical and thermal properties/variables through the measurement of any internally changing electrical property such as resistance/inductance/capacitance/phase/frequency of an actuating material under actuation. The main contribution of this paper is to obtain the stiffness from the measurement of electrical resistance of a shape memory coil during variable stiffness actuation thereby, simulating its self-sensing characteristics by developing a Support Vector Machine (SVM) regression and nonlinear regression model. Experimental evaluation of the stiffness of a passive biased shape memory coil (SMC) in antagonistic connection, for different electrical (like activation current, excitation frequency, and duty cycle) and mechanical input conditions (for example, the operating condition pre-stress) is done in terms of change in electrical resistance through the measurement of the instantaneous value. The stiffness is then calculated from force and displacement, while by this scheme it is sensed from the electrical resistance. To fulfill the deficiency of a dedicated physical stiffness sensor, self-sensing stiffness by a Soft Sensor (equivalently SVM) is a boon for variable stiffness actuation. A simple and well-proven voltage division method is used for indirect stiffness sensing; wherein, voltages across the shape memory coil and series resistance provide the electrical resistance. The predicted stiffness of SVM matches well with the experimental stiffness and this is validated by evaluating the performances such as root mean squared error (RMSE), the goodness of fit and correlation coefficient. This self-sensing variable stiffness actuation (SSVSA) provides several advantages in applications of SMA: sensor-less systems, miniaturized systems, simplified control systems and possible stiffness feedback control.

## 1. Introduction

The shape memory coil (SMC) has a larger change in force and controllable stiffness to introduce the structural elastic deformation and to be in tune with a structural load. It provides the actuation to a mechanical structure; actuation with variable load can be sensed by self-sensing the stiffness in the structure via the Shape Memory Alloy (SMA) coil’s resistance. This is because of the shape memory effect phenomenon, which is an inherent property present in the nickel-titanium alloy. This inherent property is due to phase transformation from martensite to austenite and vice-versa when subjected to temperature or current. Though the SMA has same chemical composition, atomic weight and mass number, but it is different structure in the austenite and martensite phase.

Until now, none is dedicated to physical sensor or analytical models to sense the stiffness of shape memory alloy or SMA with mechanical structure. Nowadays, the sensing function of SMA becomes more important because the high technology for the humanoid robot, industrial automation and medical field is being progressed a lot. It has been found 30 years ago that the electrical resistance of SMA changes during phase transformation of its material [1]. The modelling of SMA resistance to stiffness as compared to the models of temperature to stiffness is very rare in literature. A linear equation only is suggested between stiffness and normalized resistance. Thus, as a need of sensing/measurement and control of stiffness accurately, it must go into details of the topic. In this case, to attenuate the surrounding environmental effect, a robust and simple adaptive control is adopted. The experimental result proved that the stiffness as well as position could be controlled to achieve the desired displacement. The stiffness of the alloy varies depending on its phase. The phase of the alloy can be estimated by measuring its electrical resistance. Electrical resistance is relatively higher in martensite and lower in the austenite phase. Furthermore, a new scheme of stiffness is implemented by considering two feedback inputs- electrical resistance and position [1]. 

This electrical property is useful for sensing of thermal and mechanical properties like temperature, force and strain of SMA. Since then, many research papers have proved that resistance change is sufficiently linear to measure and control displacement and strain [2]. These works intuitively explained about the relationship between displacements in the SMA wire and its resistance. Another important point is that the SMA actuator exhibits highly nonlinear behavior. Therefore, a neural network is employed to estimate the value of displacement in the SMA from its resistance change. The estimated value of displacement from resistance is used as feedback to control it [2]. The estimation of the state (contraction/displacement) of the SMA wire actuator is done with the help of its resistance. The state of the SMA wire model has been developed by using the concept of unscented Kalman filter (UKF) which uses the measured resistance. The results are compared with the work of the extended Kalman filter and show good accuracy [3]. Accurate self-sensing concept to control the flexures by controlling the SMA wire has been also developed. Then the polynomial model is used to estimate the strain value from measurement of resistance. In addition, the inaccuracies due to the presence of the hysteresis has been overcome by pretension force. By considering standard test signals such as step and sinusoidal signals, performance of the control system has been also tested [4]. 

The polynomial model accurately enables estimation of the SMA actuator strain by applying an electrical potential across it. The experimental results have shown that the self-sensing model can achieves a small transient error and works effectively. The self-sensing helps to develop to miniaturized devices to perform effectively and efficiently [5] in which a self-sensing concept for control has been well described by adopting an antagonistic SMA wire drive. This drive is tested under different conditions such as pre-strain and duty cycles. Then, the modeling of strain and resistance is derived with the help of a curve-fitted polynomial. It has been also realized that the accurate control of an actuator wire by self-sensing feedback with a hysteresis compensator can be done [6].

From the results of this work, it has been realized that the use of neural networks to characterize the relation between the resistance of SMA wire and its strain is very effective. The great advantage of this concept is that the single SMA wire performs dual tasks as both an actuator and a sensor. This has got great importance when the prime objectives are to reduce the overall weight, size and the cost of the actuator system [7]. Artificial neural network (ANN) is applicable to accurately establish model to develop the relationship between the electrical resistance and manipulator positions. Thus, ANN can estimate the rotary manipulator position accurately. It can be controlled by the variable structure control technique under different conditions. The effect of surrounding temperature on the ability of ANN to predict the manipulator position is thoroughly investigated [8]. 

It exhibits robust performance with a small tolerance and can operate without being affected by ambient temperature. In the investigation, the authors have suggested an innovative way to calculate the resistance to determine the change in length of a SMA wire. By doing this, it is possible to measure both the voltage across the entire NiTi wire and that of the fixed-length segment. These two voltages provide direct change in length of the SMA wire. This kind of sensing is used in the feedback control in unknown ambient and loading conditions. This technique is called dual measurement for self-sensing of displacement by resistance change [9]. A self-sensing technique to measure the induced force in SMA wire is developed to control the length. Therefore, it can replace the traditional load cell by the SMA wire which can work for dual purposes as actuator and sensor. A modeling technic of the SMA wire actuator is also investigated for the control of mechanical structures. While designing a controller for the motion of a mechanical structure, a dynamic model of SMA actuator may be needed. So, the relationship between resistance and displacement of SMA that is derived to determine the feasibility of self-sensing in actuator control is investigated in [10]. The stiffness is related to force and displacement linearly as well as nonlinearly according to Hooke’s law. This web portal provides information about the basics of stiffness [11]. The physical, mechanical, electrical, and chemical properties are available on the web portal of Dynalloy Inc., and the portal is very helpful for calculations [12]. 

The sensing of displacement and stiffness without an external sensor is well described in [13]. The works [13] showed little resistance, martensite fraction, and stiffness, but not in depth like the effect of current, frequency, and pre-stress on self-sensing of the stiffness of SMA spring actuator. This enables the sensing of force without a force sensor. The direct stiffness control of the SMA actuator and sensor-less force sensing experiment is conducted successfully. Besides, several benefits of this method include simplicity of mechanism, cleanliness, silent operation, and distributed actuation system i.e., remotability, sensing ability and low driving voltage. The polynomial model of sensing the stiffness of SMA is implemented to see the influence of different activation currents and excitation/switching frequencies. It has been also shown that different activation current and excitation/switching frequencies of power transistors affected to the stiffness resistance characteristics. These experimental modeling and analyses have proved that stiffness and resistance have a sufficiently linear relation which can be easily utilized to control stiffness in a SMA spring actuator and in mechanical structures [14]. The work [14] studied the stiffness and resistance relationship with only two effects: different activation currents and switching frequencies but not the duty cycle and pre-stress. The work presented by [15,16] did not explain about the self-sensing phenomenon/concept of SMA but described modeling of stiffness-temperature and displacement with other parameters like current, temperature, resistance, and force.

A new mathematical function/model of stiffness that shows the hysteresis characteristics between the stiffness and temperature has been developed and its hysteresis characteristics between the stiffness and temperature is verified by experimental data [15]. It has been found that the hysteresis characteristics are affected by different electrical and mechanical parameters such as current, frequency, and pre-stress, respectively. The relation between stiffness and temperature in SMA spring hysteresis is experimentally verified. The hysteresis characteristics’ width and height can be controlled by current, frequency and pre-stress. 

As the SMA is a highly nonlinear element, its displacement/contraction changes nonlinearly with temperature. It is affected by many electrical and mechanical parameters. Hence, the modeling of the displacement in SMA spring actuator is important. The neural network is the best tool that can easily map one property with others. Therefore, the displacement of the SMA spring can be modeled by ANN and successfully verified by experimental data [16]. In this work, a systematic approach for the implementation of curve fitting models and methods is suggested to achieve an equation that precisely describes the sensor function [17].

It is known that the Support Vector Regression (SVR) technique is normally applied to forecast the tangential displacement of cement concrete dam. Thus, in general, it is tested and verified using Pearson correlation coefficient, mean absolute error (MAE) and mean squared error (MSE) with experimental data [18]. The implementation of SVR is a practical and user-friendly method for creating soft sensors for nonlinear systems. The development of a dynamic non-linear-Auto Regressive (ARX) model-based soft sensor employing SVR is suggested as a heuristic method, in which the ideal delay and order are determined automatically using the input-output data. It is noted here that an Online Support Vector Regressor (OSVR) model is effective to estimate chemical process variable. As mentioned earlier, many works investigated the self-sensing technique to relate position/length/displacement with resistance and its control. The soft sensor developed using SVR in [18,19,20] achieved excellent performance checked by statistical performance parameters such as MAE, MSE and correlation coefficient. The research work [21] gives the idea about the stiffness of shape memory coil in terms of resistivity and modulus of elasticity. It suggests the resistivity and shear modulus is the best alternative to existing self-sensing methods. 

In research article [22], the overall stiffness is adjusted by modifying the shape of the leaf springs. Hence, the geometrical nonlinearity can be used to change global stiffness. The paper [23] method suggested in adapting stiffness in variable stiffness actuator by configuring the fluid circuits, while the humanoid robot is investigated in [24]. It has various interesting features and is a difficult mechatronics structure. Due to the close interdependence of the technological factors, it is challenging to conduct research in a specific direction. A parallel type SMA wire variable stiffness actuator with a synergistically constructed configuration that offers a small size and a wide range of stiffness adjustments in compliant structures is also studied [25]. Additionally, it provides sufficient displacement and force, making it appropriate for applications requiring peculiar soft robotic requirements. The research work [26] proposes a new type of pneumatic variable stiffness actuator (PVSA). It provides expected actuation performance with effective remote stiffness adjustment capability. A novel variable stiffness mechanism is also designed by using specially designed SMA S-spring with different thickness [27]. The actuator stiffness is discretely adjusted by changing the state combination of SMA S-spring with different thickness. The [28] work is self- sensing unique design of sandwich structure comprising active graphene coated glass fabric piezoresistive face sheets bonded to a Nomex™ honeycomb core. The research article [29] explains the design of SMA spring and how to improve the frequency of actuation. The research [30] demonstrates the ability to significantly improve the way a gripper interacts with things that are being handled and offers a path toward developing anthropomorphic grippers. The soft finger’s built-in sensor may convey passive proprioceptive feelings of stiffness and curvature. While not altering the mechanics of the robotic movement, it also served as an active jamming element to adjust finger stiffness.

As evident from the literature survey, the self-sensing during actuation of SMA spring is very useful to understand the relationship between the stiffness change and force, displacement, and strain corresponding change in the electrical resistance during the phase transformation. However, so far, a comprehensive study considering several effects such as actuation current, excitation frequencies and duty cycles has not been reported yet. Consequently, the technical novelties of this study are summarized as follows: (a) achievement of the self-sensing behaviour of SMA spring during variable stiffness actuation by experimentation, (b) development of a data driven model of self-sensing variable stiffness actuation based on Support Vector Machine (SVM) regression and nonlinear regression methods, by availing the experimental data, (c) investigation the effect of different excitation/activation currents, switching/excitation frequencies and duty cycle, (d) evaluation of the cycles and pre-stresses on the stiffness-resistance characteristics during the heating cycle of SMA spring by employing SVM algorithm as a soft sensor. From the aspects of the technical novelties, several new characteristics which are significant for the development of self-sensing are found. Some of new findings are given as follows: (i) it is identified that the characteristics of the self-sensing actuation (SSA) are influenced by activation currents, switching/excitation frequencies, duty cycles and pre-stresses, (ii) both SVM regression and nonlinear regression models are acceptable to measure the self-sensing stiffness in SMC actuator during variable stiffness actuation which is experimentally validated with its significance, (iii) it is found that the electrical resistance for all factors is almost in linear relation with the stiffness of SMC during the heating cycle with a meagre hysteresis, (iv) the resistance of SMC is low under the austenite and high under the martensite phases. It is noted here that compact design of a self-sensing SMA spring/wire actuator device is feasible with the aid of the results achieved in this work. 

This paper is organized as follows. After clearly describing the research motivation, literature survey and the technical contributions of this study, an experimental facility and measurement method are presented in Section 2. Section 3 provides the information about SVM regression and nonlinear regression models focusing on the usability, and the basic mathematical relations used to find the resistance and stiffness which are undertaken at various different conditions of activation/excitation current, switching/excitation frequency, duty cycle and pre-stress are given in Section 4. The detailed characteristics on the results regarding to the stiffness response, resistance response and stiffness-resistance characteristic is fully discussed in Section 5 with the brief information about application of nonlinear regression modeling, followed by conclusion in Section 6 where some of benefits achieved form this work such as robust model, cost effectiveness and reliability for compact design of self-sensing actuator.

## 2. Experimental Set Up 

### 2.1. Facility

The study to self-sense the stiffness of the SMA spring during variable stiffness actuation from resistance measurement is highly significant with respect to the quality of device (in terms of accuracy, precision, sensitivity and linearity etc., compactness and cost effectiveness). This experimental study is used to validate Support Vector Machine Regression and Nonlinear Regression model and realized by MATLAB 2020b software. A set-up to run the tests is designed and fabricated and shown in Figure 1 with the help of the photograph. It has following sections: (a) Mechanical Actuation System—It has two guide rods, with two linear bearings on them to actuate the SMA spring biased with an antagonistic tensile passive steel spring. This helps obstacle free movement which is measured with the help of a flap placed between two springs and Keyence—made contactless laser displacement sensor. A force sensor is also connected between the fixed frame and the SMA spring. The complete assembly is fixed in an acrylic frame. (b) Electronic Actuation System—This system consists of on Metal-Oxide-Semiconductor Field-Effect Transistor (MOSFET, TIP-122 ST Microelectronics) with gate resistance (1.5 kΩ and ¼ watt) to limit the base current; a source with a rheostat (4 Ω and 8.5 A) and drain is connected to the ground. The SMA spring is connected between the rheostat and the source of power transistor. The current sensor is connected between the ammeter and the rheostat so that it can display the current flowing through the SMA spring. (c) Power Supply—Different power supply systems are required for working of different auxiliary devices and the complete actuation systems. (i) DC regulated power supply—this is required for relay circuit, excitations for current sensor, transistor circuit, and as an input signal to op amp circuits. (ii) Dual Power Supply—Dual power supply provides +/− 15 V and 0.5 A current. This is important for op amp-based amplification circuits to boost physical signals like temperature signal and displacement signal. (iii) AC Power supply is required for bigger auxiliary devices. (d) Instrumentation and Data Acquisition (DAQ) System—These includes current sensor, temperature sensor, miniature force sensor, laser displacement sensor, voltages across SMA spring and rheostat.

Voltage signals from different sensors are converted into proper level (0 to 10 V) through signal conditioning, so that it is compatible with the data acquisition system and stored in the personal computer’s memory. Figure 1 also shows the DAQ card used for data acquisition and forwarding it to the computer memory. (e) Shape memory alloy spring and passive steel spring—The SMA spring is manufactured by Dynalloy Inc. (1562 Reynolds Avenue (949) 502-8548 office, Irvine, CA 92614 USA) and their technical specifications are given in Table 1. Other details of the passive steel spring are shown in Table 2. During activation/heating of the SMA spring, the biased tensile spring expands and stores the mechanical energy. During deactivation/cooling, the bias spring will use the stored energy to pull the SMA spring back to its pre-stress/deform state.

### 2.2. Experimentation

The experimental modeling and analysis of self-sensing of the stiffness in the Shape Memory Coil during variable stiffness actuation is performed in the following ways. In the first set of experiments, activation currents (0.8 A, 1.0 A and 1.2 A) are varied by D.C. regulated power supply and by keeping voltage constant. Then different sensor voltages, the voltage across rheostat and voltage across the Shape Memory Coil are recorded in the memory of personal computer via DAQ card through repetitive cycle of switching of the power transistor. This procedure is repeated for aforementioned activation currents by keeping the switching frequency, duty cycle and pre-stress constant. The recorded information is used to predict stiffness by nonlinear regression modeling and validation. It is presented in detail in Section 4.

The recorded instantaneous value is used to determine the resistance and stiffness properties of SMC by use of Equations (6) and (7) for the aforementioned activation currents. In the second set of experiments, the switching frequency of the power transistor is changed e.g., 10 mHz, 20 mHz and 30 mHz by ensuring that all other parameters such as activation current, pre-stress on the SMC and duty cycle of the switching frequency are constant. The instantaneous values of the SMA spring’s properties in terms of voltages is continuously recorded in the memory of personal computer via DAQ card. These recorded voltages are converted into proper units of the properties of SMC such as force, displacement, temperature, resistance, and stiffness.

Also, the experimental modeling and validations are explained in detail in Section 4. In the third set of experiments, the pre-stress (100 g, 150 g and 200 g) on SMC is varied by applying more tension with the help of a tensile passive steel spring. All other parameters are kept constant mentioned in the earlier set of experimentation. Similarly, the properties of SMC in terms of voltages are recorded by different sensors and described in detail in Section 4. Some of the properties of SMC e.g., stiffness and resistance are derived with help of Equations (6) and (7), respectively. In the fourth set of experiments, the duty cycle (40%, 50%, and 60%) of switching frequency is varied by adjusting the knob of the function generator such that activation and deactivation of SMC occur smoothly with help of passive bias tensile spring and explained in detail in Section 4. Basically, SMA works in three different modes of operations as actuators: (i) Free recovery mode means a constant force and variable contraction of SMC. (ii) Constraint recovery mode means a variable force and fixed contraction of SMC. (iii) Work production mode means both force and contraction of SMC changing. The work production mode is the most popular and applicable to practical engineering applications. It is a controllable actuator where both force and displacement vary. The first mode of operation is free recovery where force is constant, and stiffness is a function of displacement and varies. In the second mode, displacement is constant in which stiffness is a function of force and varies [15]. The inferior vena cava filter and eyeglass frame are designed in the free recovery mode of SMA operation. Fasteners, connectors, and hydraulic coupling uses constrained recovery mode in the SMA operation. Circuit breakers, heat engine and actuators are a few applications of the work production mode in the SMA operation [15]. In the experiment, the variable stiffness actuation of the SMA coil is controlled by using currents of 0.8 A, 1.0 A, and 1.2 A, and the corresponding forces, displacements, currents, and voltages are monitored. The data are displayed to investigate the system characteristics after preprocessing.

## 3. Facility and Experimentation

### 3.1. Principle of Self Sensing of Variable Stiffness Actuation by Support Vector Machine Regression

The regression is statistical tool to model and analyze the relation between one or more than one independent variable and a dependent variable to produce a particular outcome. In other words, the basic idea is to approximate the functional relation between set of independent variable and dependent variable by minimizing risk function which will use prediction error. Kernel: A lower dimensional data set is mapped into a higher dimensional data set using the Kernel function. In essence, a hyperplane is a line that will enable us to estimate a continuous value or aim. Boundary Line: In SVM, a margin is created by two lines other than the hyperplane. The Support Vectors may be on or outside the boundary lines. The experimental data recorded for resistance and stiffness of shape memory coil are denoted as $X_{i}$ and $Y_{i}$.
(1)$$\{\;(X_{1},Y_{1}),\;(X_{2},Y_{2}),(X_{3},Y_{3}),\ldots \ldots \ldots ..(X_{n},Y_{n})\}\;\in R^{N}\times R\;$$
where, *i* is varying from 0 to *n* and *n* is number of training data points. The goal is to find the function which map the relation between $X_{i}$ and $Y_{i}$. The Support Vector Regressor algorithm approximate the function as,
(2)$$f\left(x\right)=<w,\varphi \left(x\right)>+b\;\;and\;w\;\in R^{N},\;b\;\in R$$

The one dimensional and multidimensional SVR problem is defined as
(3)$$f\left(x\right)=\sum_{j=1}^{M}w_{j}*x_{j}+b\;\;and\;w\;\in R^{M},\;b\;\in R$$
(4)$$f\left(x\right)=\left(\begin{array}{c} w\\ b \end{array}\right)\left(\begin{array}{c} x\\ 1 \end{array}\right)=w^{T}\;x+b;\;x,w\in R^{M+1}$$
where, *w* is weight vector, *b* is bias and *φ(x)* is high dimensional data space. The weight vector and bias can be determined from risk function and as,
(5)$$R\left(C\right)=\frac{1}{2}|\;w|{\;}^{2}+C\frac{1}{2}\sum_{i=1}^{n}L_{ɛ}\left(f\left(X_{i}\right),\;Y_{i}\right)$$

The $\frac{1}{2}|\;w|{\;}^{2}$ control the function capacity and $C\frac{1}{2}\sum_{i=1}^{n}L_{ɛ}\left(f\left(X_{i}\right),\;Y_{i}\right)$ is the error. *C* is regularization constant. The insensitive loss function is defined as,
(6)$$L_{ɛ}\left(f\left(X_{i}\right),\;Y_{i}\right)=\left\{\begin{array}{cc} |f\left(X_{i}\right),-Y_{i}|-ɛ. & \;\;when\;\;\;|f\left(X_{i}\right),-Y_{i}|\geq ɛ\;\\ 0\;, & and\;otherwise\; \end{array}\right.$$
where, ɛ is the boundary line [18].
(7)$$f\left(X_{i}\right),=\varphi {\left(x_{i}\right)}^{T}w+b$$

*φ(x)* is high dimensional data space.

### 3.2. Principle of Self-Sensing of Variable Stiffness Actuation by Nonlinear Regression

Most self-sensing actuation model literature have related SMA in terms of the displacement/strain with self-sensed electrical resistance. To represent the stiffness of SMA in terms of its electrical resistance, there is a need to establish an appropriate and reliable stiffness-resistance model. To find the appropriate mathematical model that expresses the relationship between dependent variable (stiffness) and the independent variable (resistance), a data driven model is used; it is a parallel to mathematical model with self-sensing characteristics i.e., sensing under actuation. So, the data driven model is a function of the independent variable involving one or more coefficients. The nonlinear regression model with continuous one-to-one mapping is efficient to describe the relation between the dependent variable (stiffness) and independent variable (resistance). The Nonlinear Regression model is preferred as the modeling of the self-sensing actuation phenomenon (the relation of change in electrical resistance to the change in stiffness during the nonlinear thermo-mechanical phase transformation) is not as complex as that of modeling the basic phenomenon of SMA, the thermo-mechanical phase transformation. Also, the Nonlinear Regression model can be used to approximate complex nonlinear phenomena and then the relationship is curvilinear. The *j*th order polynomial model in one variable is given by,
(8)$$k=\beta_{0}+\beta_{1}R_{sma}+\beta_{2}R_{sma}{}^{2}+\ldots +\beta_{j}R_{sma}{}^{j}+\varepsilon$$
where, $R_{sma}$ is resistance of SMA spring, *β*_0_, *β*_1,_
*β*_2_,..., *β_i_*, and $R_{sma}\;R_{sma}2\;R_{sma}3$*…*$R_{sma}$ *…*$R_{sma}i$, *i* = 1, 2, 3,…, *j* are the effect parameters and ε an error.

The nonlinear regression of sufficiently high degree can always be found that provides a good fit for data. A good strategy should be used to choose the order of an approximate polynomial; keep the order increasing until t-test for the highest order term is non-significant. It is called the forward selection procedure to fit the model with experimental data. Also, goodness of fit statistics is used to find the best polynomial. MATLAB function “polyfit” is used to obtain coefficients of the polynomials [3]. Data from the measurement of the force, displacement and voltage sensors is saved in an Excel.csv file and used whenever required for training and testing the model. In the first step, characteristics must be represented as predicted data, response data, and weights. In the second step, nature i.e., shape and specificity of the self-sensing characteristic of the appropriate parameter polynomial model is selected. After fitting the data into a model, its goodness of fit is determined by adopting any of the following two ways: (i) Graphical (ii) Numerical. The plotting residuals and prediction bound aid visual interpretation. Graphical measures allow viewing the entire data set at once, and they easily display a wide range of relationships between the model and data. Numerical measures are more narrowly focused on a particular aspect of the data and often try to compress that information into a single number. In practice, to find the best fit of the sensor’s characteristics, the above-mentioned methods are used on extensive experimental data and analyzed [17]. Figure 2 shows the trial-and-error procedure to find the correct polynomial model in comparison with the experimental sensor’s characteristics; it can be seen that the third-order and above models match the experimental data.

## 4. Result and Discussion of Self-Sensing Variable Stiffness Actuation

The influence of different factors such as activation current, excitation frequency, duty cycle, and pre-stress on self-sensing characteristics are observed and presented in this section. The influence of different factors such as activation current, excitation frequency, duty cycle, and pre-stress on self-sensing characteristics are observed and presented in this section. The SMA’s electrical resistance is sensitive to compositions, transformation path and heat treatment [1]. During phase transformation, the crystallographic structure of SMA changes due to heating and cooling. As a result, SMA’s electrical resistance changes. This resistance change is useful to measure SMA’s stiffness without any physical sensor. Furthermore, it is possible to measure stiffness of the SMA based structure. Table 3 has useful information about SMA material. It is provided by the manufacturer and that the resistance of the SMA wire depends on its length and diameter. It also says that resistance per meter decreases with an increase in wire diameter. The basic relation used to calculate resistance [3,5] and stiffness of the SMC actuator are as follows.
(9)$$R_{\mathrm{SMA}}=\frac{V_{\mathrm{SMA}}}{Vs-V_{SMA}}*\;R\;$$
(10)$$k=\frac{d^{4}G}{8\;n\;D^{3}}$$
where, *R_SMA_* is the resistance of the SMA spring (Ω), *V_SMA_* is the voltage of the SMA spring (V), *R* is the known resistance (Ω), *V_S_* is the bias voltage of the MOSFET (V),

*k* is the instantaneous stiffness of the SMA spring (N/m), *G* is the instantaneous shear modulus of the SMC (N/m^2^), *d* is the wire diameter of the SMC (m), and *D* is the coil diameter of the SMC (m). The instantaneous value of *G* is calculated from force and displacement measurements using transducers.

The first step in the design process is to choose the smallest wire diameter, or “*d*”. The force-displacement relationship and cooling performance associated to the actuation frequency are most sensitively influenced by the wire diameter “*d*”, which is a dominant design parameter of the SMA coil spring actuator. The SMA coil spring actuator has the smallest material mass and the quickest cooling time when the wire diameter is the smallest.

Iterative calculations are used to determine the coil diameter “*D*”. When the shear strain reaches the predetermined value, which is adjusted to be slightly greater than the wire diameter “*d*”, the force is calculated. If “*D*” is tiny, the shear stain does not reach its maximum value before the force surpasses the intended value. If so, the calculation is redone with a larger “*D*” The iteration ends and the “*D*” value at the last step is the maximum coil diameter if the force reaches the required value at the maximum shear strain while “*D*” is growing. With the desired stroke and the single coil stroke at the necessary loading condition, the coil number “*n*” is calculated. The displacement gap between the martensite and austenite models is used to determine the single coil stroke. The desired actuation stroke value divided by the single coil stroke yields the coil number “*n*”.

So, the initial value of displacement of SMC is assumed to be zero and it is set to zero in the transducers and recorded in the personal computer. This modeling is aimed at the self-sensing phenomenon when the SMA is under variable stiffness actuation i.e., not particularly on the modeling of the basic actuation or shape memory effect or phase transformation of SMA. Moreover, the study is based on the SMA being activated by joule heating, whereby it is controlled by different electrical parameters such as current, frequency and duty cycle and, pre-stress. The change in resistance corresponding to the change in stiffness is determined during the heating phase, and the relation is extracted as a nonlinear regression model and validated by different metrics through experimentation. The model is valid for the joule heating current of 0.7 A to 1.2 A. Figure 3 shows the stiffness characteristics at 0.8 A, 1.0 A and 1.2 A which is when the phase change starts before which the SMA does not display any linear response. But then after 1.5 A, the response is more linear in characteristic as seen in Figure 3 and also at a current 1.0 A and 1.2 A, respectively. The model and experimental response are compared in the heating cycle when work is completed, specifically between 0.7 A and 1.2 A also, wherein a large change in stiffness and resistance is revealed.

### 4.1. The Effect of Different Activation Currents

In the first set of experiments, the SMA spring is electrically heated by a 1/100 Hz square wave signal with a constant duty cycle and constant pre-stress. The current of the heating signal is varied, and the data recorded. The current/electrical power affects both the stiffness and resistance of the SMA spring actuator. The effect of changing current is clearly seen in Figure 3. Stiffness-resistance heating characteristics are modeled by the SVR and Nonlinear regression. Figure 3 reveal, that as electrical current increases, the slope of the curve increases, and the experimental characteristics overlap the modeled characteristics. The implementation of both models is done in MATLAB by “polyfit”, “polyval” and other built-in functions.

The mathematical Nonlinear regression model between stiffness and resistance during the austenite phase is estimated from the experimental data.
(11)$$k=-1.6446*{R_{SMA}}^{3}+2.7072*{R_{SMA}}^{2}-2.0349*R_{SMA}+0.9655$$
(12)$$k=-0.5627*{R_{SMA}}^{3}-0.0684*{R_{SMA}}^{2}-0.3944*R_{SMA}+1.0032$$
(13)$$k=1.1301*{R_{SMA}}^{3}-0.0543*{R_{SMA}}^{2}-1.9872*R_{SMA}+1.0067$$
where, *k* is the instantaneous stiffness in N/m and “*RSMA*” is the resistance in ohm of the SMA spring actuator. Three cases are considered to present the data uniformly corresponding to each effect in parameter variations like current, frequency, duty cycle and pre-stress. The Nonlinear Regression model is used to represent self-sensing actuation in particular to relate stiffness with electrical resistance. The big and unusual coefficient of the nonlinear regression model can be reduced by normalizing stiffness and resistance data. The modeled and experimental self-sensing of stiffness of the Shape Memory Coil during variable stiffness actuation agree in terms of quality as Figure 3 has performance metrics such as goodness factor, mean squared error, correlation matrix and root mean square error within the specified range. The root mean square error (RMSE) should be less than 0.80. The goodness factor is out of range in Figure 3 as nonlinearity is present for the phase conversion which has not yet started.

The values of metrics of the model comparison for Figure 3 is mentioned in Table 4; there is large difference in stiffness for the three (maximum values are 75, 145 and 1500 N/m) cases. Table 5 gives the Correlation matrix which displays the correlation coefficients for matching the stiffness (model and experimental) at different independent variables, and the activation current. The matrix depicts the correlation between the possible pairs of values in the table; this tool helps to summarize the dataset, to identify and visualize the match of patterns in the data. From the matrix tables it is observed that, the characteristics at 0.8 A do not match as they are more non-linear than that for the other two higher activation currents.

### 4.2. The Effect of Different Excitation Frequencies

In the second set of experiments, the Shape Memory Coil is electrically heated over a fixed current of 1.2 A of different frequencies (10 mHz, 20 mHz, and 30 mHz) of square wave signal with fixed duty cycle and pre-stress (pre-tension). The self-sensing of stiffness is modeled during the heating cycle only. Both the curves almost agree with each other (modeled and experimental). Figure 4 show the characteristics modeled and experimental plots at different frequencies. The effect of frequencies on stiffness is inversely proportional i.e., stiffness decreases when resistance increase with an increase in frequency. The quadratic mathematical models of stiffness at different frequencies are as follows: (14)$$k=1.1074*{R_{SMA}}^{2}-1.9230*R_{SMA}+0.8835$$
(15)$$\;k=1.5465*{R_{SMA}}^{2}-2.4596*R_{SMA}+0.9701$$
(16)$$k=-0.5693*{R_{SMA}}^{2}-0.2908*R_{SMA}+0.9574$$

The experimental results validated the self-sensing of stiffness of the SMC actuator by measurement of resistance during the heating cycle (austenite phase). Figure 4 reveals the linear relationship between these two properties of the SMA. Also, the resistance change of the SMA spring actuator is higher at a lower frequency and lower at higher frequency over 0 to 1.2 A of electrical power. 

The modeled and experimental self-sensing of stiffness of the SMC during variable stiffness actuation agree in terms of quality because Figure 4 has the performance metrics such as goodness factor, standard deviation, correlation matrix and root mean square error within the specified range, as seen from Table 6. Also, Table 7 provides the correlation matrix which validates the match for the range of frequency (10 mHz to 30 mHz) though the study is conducted until 0.6 Hz.

### 4.3. The Effect of Different Duty Cycles

An SVR and Nonlinear regression model is developed to understand the effect of duty cycle on stiffness-resistance characteristics. The comparison of experimental and curve-fitted model is simulated by the MATLAB^®^ program, the characteristics for different duty cycles (40%, 50% and 60%), at a constant current of 1.5 A and constant frequency of 20 mHz showed that they almost agree with each other. The nonlinear regression model of stiffness at different duty cycles with resistance change as independent variable are as follows:(17)$$k=0.5778*{R_{SMA}}^{2}-1.5332*R_{SMA}+0.9982$$
(18)$$k=0.3808*{R_{SMA}}^{2}-1.3471*R_{SMA}+1.0264$$
(19)$$k=-0.2699*{R_{SMA}}^{2}-0.7399*R_{SMA}+1.0239$$

The effect of duty cycle on stiffness-resistance characteristics are linear and useful in controlling stiffness effectively. The comparison between the experimental and modeled characteristics at different duty cycles are presented in Figure 5 and mathematically represented by quadratic (second-order polynomial) Equations (17)–(19). The modeled and the experimental self-sensing of stiffness of the SMC during variable stiffness actuation agree in terms of quality as Figure 5 contains performance metrics such as goodness factor, mean squared error, correlation matrix and root mean square error within the specified range, as seen in Table 8. Table 9 also gives the correlation matrix which validates the match for a range of duty cycle (40% to 60%), though the study is conducted from 20% to 80%.

### 4.4. The Effect of Different Pre-Stresses (Pre-Tension)

The effect of pre-stress on stiffness-resistance characteristics is developed as mathematical models for different pre-stresses (100 g, 150 g and 200 g) at a constant current of 1.2 A and constant frequency of 10 mHz. The nonlinear regression model of stiffness at different duty cycles with resistance change as independent variable are as follows:(20)$$k=-2.9194*{R_{SMA}}^{3}+5.3974*{R_{SMA}}^{2}-3.3947*R_{SMA}+0.8517$$
(21)$$k=-2.4860*{R_{SMA}}^{3}+4.6766*{R_{SMA}}^{2}-3.0950*R_{SMA}+0.8585$$
(22)$$k=-0.9385*{R_{SMA}}^{3}+1.5918*{R_{SMA}}^{2}-1.5918*R_{SMA}+1.0084$$

The comparison between experimental and modeled characteristics at different pre-stresses found in Figure 6 and mathematically represented by third order polynomial Equations (20)–(22). The effect of pre-stress on stiffness-resistance characteristics is highly nonlinear and difficult to model in comparison with those on the effect of current, frequency and duty cycle. Figure 6 reveals this and it is found from the correlation matrix that stiffness and resistance of the SMA spring at different stresses do not have strong statistical correlation. Table 10 validates the two models, compares the two models with each other and gives the information about the accuracy of prediction by using experimental and predicted data with the help of different metrics. There is a perfect correlation of each variable with itself as seen from Table 11. 

## 5. Investigation of Stiffness Characteristics of the SMC Actuator

### 5.1. Effect of Current on Stiffness-Resistance Characteristics

With the help of four sets of experiments, the stiffness-resistance characteristics of the SMC actuator are analyzed to explore its self-sensing capability. Data recorded from the first set of experiments are used to plot and analyze. Resistance response to the heating cycle is plotted for 50 s; it found that as activation current increased, resistance decreased and that the change of resistance decreased over a period of time as shown in Figure 7. Stiffness response to different activation currents found that at 1.2 A, stiffness increased very rapidly in comparison to the other two activation currents. Excitation frequency is chosen as 10 mHz, as enough time is available to relax/deform the SMA spring and avoid residual strain. As excitation frequency increased, the time to complete the cycle is reduced e.g., two heating-cooling cycles occurred at 10 mHz and 6 cycles at 50 mHz. Data of force, displacement, the voltage across fixed resistance and the SMA spring are recorded for 3 min and are also saved in computer memory via 1408FS plus DAQ card with a sampling frequency of 2 Hz. At 0.8 A stiffness is less in value, compared to the other two activation currents (1.0 A and 1.2 A) as the SMA spring does not completely transform from martensite phase to the austenite phase. Corresponding resistance (Ω) is recorded in terms of its voltage and plotted in Figure 7; at 0.8 A. It is observed that resistance change is larger than that at 1.2 A. Responses for the three different currents are plotted and shown in Figure 8 and Figure 9. Figure 8 shows stiffness variations at different currents (0.8 A, 1.0 A and 1.2 A). Figure 9 shows a linear relation between stiffness and resistance. The minimum and maximum values of resistance at these activation currents and their respective stiffness values are shown in Table 12.

### 5.2. Effect of Frequency on Stiffness—Resistance Characteristics

Data recorded from the second set of experiments are used to plot and analyze. Self-sensing actuation characteristics of the SMA spring actuator is obtained for varied frequencies from 10 mHz to 30 mHz keeping activation current constant at 1.2 A; Stiffness is determined and plotted as shown in Figure 10, Figure 11 and Figure 12. When the excitation frequency of PWM signal is increased beyond 50 mHz, the number of cycles is reduced to less than one, also the heating and cooling cycle frequency (mechanical cycle) did not match the excitation frequency (electrical cycle), subsequently, the SMA spring would not completely contract or deform. Some significant observations are arrived at from these plots: (i) The excitation frequency has a significant effect on stiffness and resistance of Shape memory Spring: At a higher frequency, resistance change is higher and at a lower frequency, the resistance change is lower. (ii) The effect of frequency on stiffness change of the SMA spring actuator is converse to resistance change. (iii) Overall linear relationship exists between resistance and stiffness. (iv) The Resistance and Stiffness change from minimum to maximum values at different frequencies is in Table 13.

### 5.3. Effect of Pre-Stress on Stiffness—Resistance Characteristics 

At a constant current (1.2 A) passing through the SMA spring and constant frequency (10 mHz) of excitation current, stiffness is determined for pre-stress which is varied from 100 g to 200 g. An increase in pre-stress beyond 200 g would not allow complete contraction (bias force is higher) and would not completely deform below 100 g for the requirement of restraining/pulling force [15]. It is learnt from Figure 13 and Figure 14 that stiffness increased, and resistance decreased with an increase in pre-stress. Figure 15 provides information about the stiffness sensing characteristics at different pre-stresses, which reveal a large change in stiffness at a lower value of resistance of the SMA spring actuator.

### 5.4. Effect of Duty Cycle on Stiffness—Resistance characteristics

Similarly, for different duty cycles (40%, 50%, and 60%) at 1.5 A and 20 mHz, the resistance response, stiffness response and stiffness—Resistance characteristics of the SMA spring actuator are obtained and presented in Figure 16, Figure 17 and Figure 18 respectively. Resistance values are smaller in comparison to the effect of current, frequency and pre-stress. The change in resistance is higher for higher duty cycles and lower for lower-duty cycles. Change in stiffness is also higher for a higher duty cycle and lower for a lower duty cycle; stiffness is higher in comparison with a lower duty cycle due to the availability of minimal time to completely heat the SMA spring. Figure 16 presents the resistance variation due to changes in duty cycles and Figure 17 corresponds to the stiffness variations. As the duty cycle increased, stiffness also increased some extent. Figure 18 shows the characteristics at different duty cycles, indicating a large change in stiffness with a large change in the resistance of the SMA spring actuator at a higher duty cycle. It also, proved that at higher stiffness, resistance is also higher but within the specified limit of duty cycle. Table 14 summarizes stiffness and resistances at different pre-stresses and their effect. Table 15 depicts the effect of duty cycle on stiffness and resistance, which is more on stiffness and less on resistance.

Experimental data is collected for repeated cycles (3 consecutive cycles), and the responses are similar/not with much deviation. One set of data is used for the plotting and analysis of the stiffness-resistance characteristics, the usable form of the data is through the use of an average function and normalization; the other set of data are used for the validation. The highest change in resistance corresponding to the change of stiffness with regard to the effect of the influencing factors like activation current, excitation frequency, pre-stress and duty cycle are presented in Table 16. It is observed that a change in resistance of the SMA spring in the configuration switches is the highest (0.3516 Ω) during the change in pre-stress and stayed the lowest (0.0160 Ω) during the change in duty cycle. The purpose of this study is to use the suggested technique during control of actuation in the SMA spring by using an appropriate controller. The polynomial models are appropriate to the relevance to the factor of consideration and are able to accurately predict the stiffness equivalent to that obtained through experimentation by the measurement of change in electrical resistance. The level of predictability is high for the factor’s activation current, excitation frequency and duty cycle but low for pre-stress and low value of current (0.8 A) due to nonlinear characteristics of self-sensing of the SMA spring; the statistical analysis is presented in Table 4, Table 5, Table 6, Table 7, Table 8, Table 9, Table 10 and Table 11. Table 17 attests that the polynomial model, predicted accurately at different activation currents, excitation frequencies and duty cycle but at the pre-stress. 

## 6. Conclusions

In this work, an experimental facility is developed to determine the electrical resistance of the Shape Memory Coil (SMC) actuator that is biased by a tensile steel spring under self-sensing variable stiffness actuation. The SVM regression model is constructed based on experimental data (Expert Knowledge) and provided excellent performances. The performance of the SVM regressor model is verified by a correlation coefficient, mean square error (MSE), root mean square error (RMSE) and goodness of fit (R^2^). The developed SVM model showed an excellent result of prediction in comparison with the nonlinear regression model and experimental data. The experimental analysis has proved that the stiffness of the SMA is sensed from its resistance change. While the stiffness is changed due to different activation currents/joule heating, excitation frequencies, pre-stresses and duty cycles, the stiffness of the SMC is successfully determined as the variable stiffness actuator. Among many new findings from this work, the most interesting result is that the stiffness of the SMA spring can be measured without knowing the activation current and initial geometry or configurations of the SMC. This is possible from the realization of both SVR and nonlinear regression models of the stiffness using the electrical resistance of the SMC during austenite phase transformation. The responses achieved from two models are compared to the experimental response showing both models would harness the self-sensing capability of the SMC actuator. In addition, it has found from this work that the effect of frequency and duty cycle is more linear when compared to the other two parameters of current and pre-stresses. It has been concluded from this work that from the practical view of point, the self-sensing of stiffness of SMC can reduce the number of sensors for making application systems associated with shape memory alloys. For example, one stiffness sensor can use for two sensors of force and displacement. Therefore, it is self-explanatory justifying that the self-sensing technique gives birth to sensor-less control systems which are relatively cost effective, and hence the overall system becomes compact in comparison with traditional control systems having dedicated many sensors. Therefore, it is expected that the proposed self-sensing variable stiffness actuation method can be applicable to many control systems including the grasping force of robot grippers, surgical SMA wire of biomedical sciences, vibration and dynamic motion control of flexible structures in aeronautical fields such as morphing control and health monitoring control system using SMA wires Associated magnetic coils. It is finally remarked that some benefits achieved from this work will be demonstrated by applying to robot gripper systems in near future. 

## Figures and Tables

**Figure 1 sensors-23-02442-f001:**
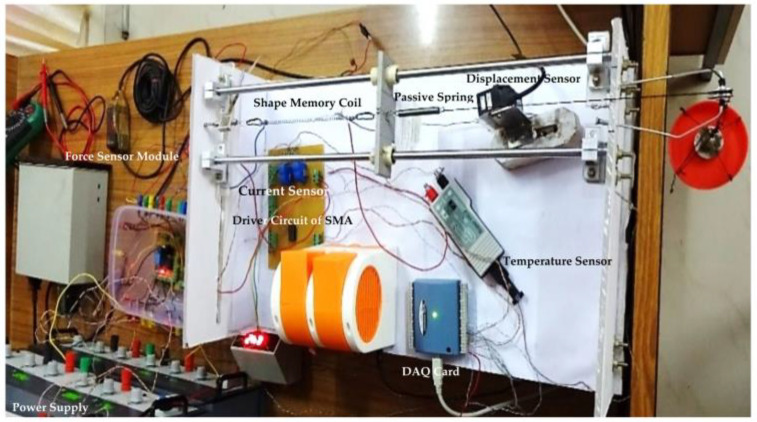
Experimental setup.

**Figure 2 sensors-23-02442-f002:**
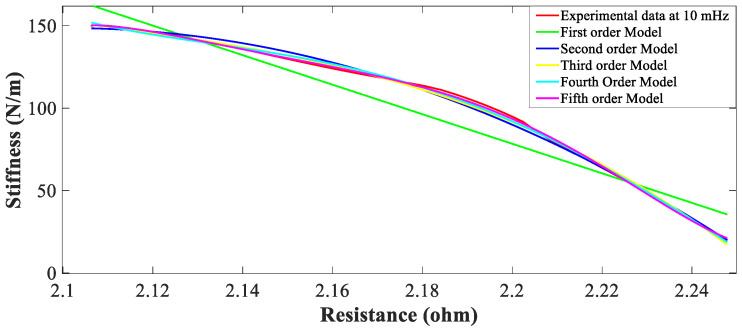
Polynomial model and experimental self-sensing variable stiffness actuation characteristics.

**Figure 3 sensors-23-02442-f003:**
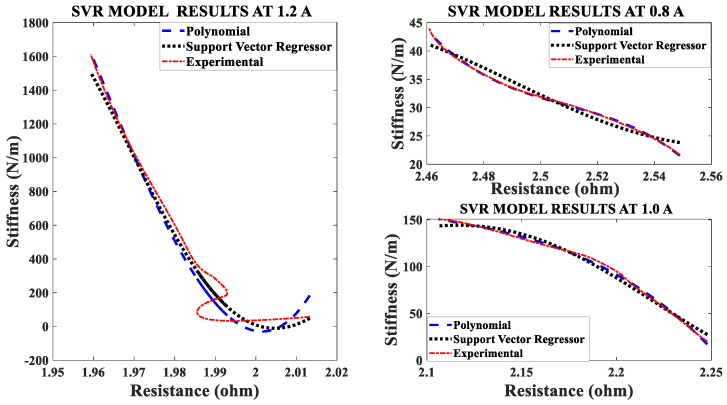
Comparison of models of stiffness sensing characteristics at different currents.

**Figure 4 sensors-23-02442-f004:**
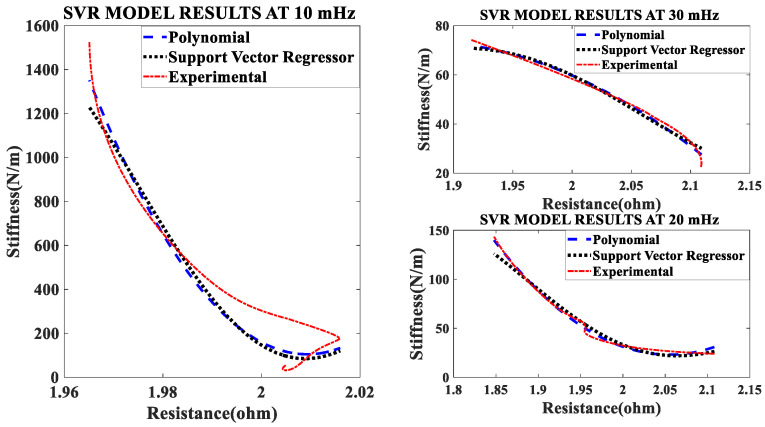
Comparison of models of self-sensing stiffness characteristics with experimental result at different excitation frequencies.

**Figure 5 sensors-23-02442-f005:**
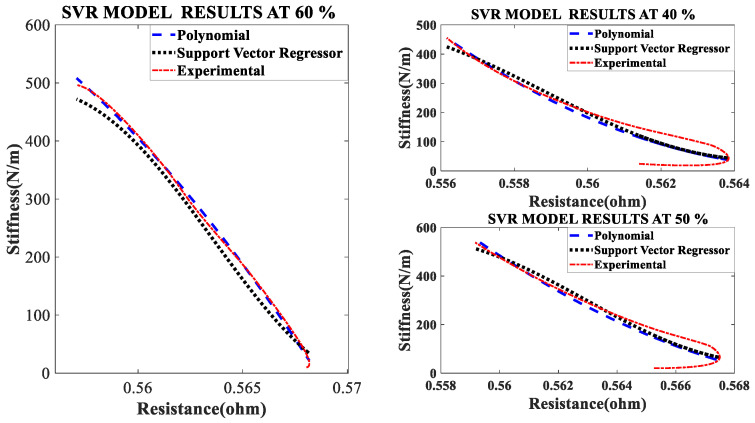
Comparison of models of self-sensing stiffness characteristics with experimental results at different duty cycles.

**Figure 6 sensors-23-02442-f006:**
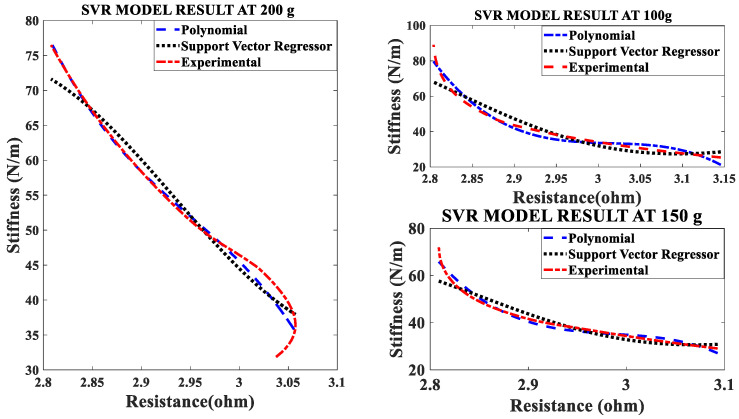
Comparison of Models of Self-sensing Stiffness characteristics with experimental results at different pre-stresses.

**Figure 7 sensors-23-02442-f007:**
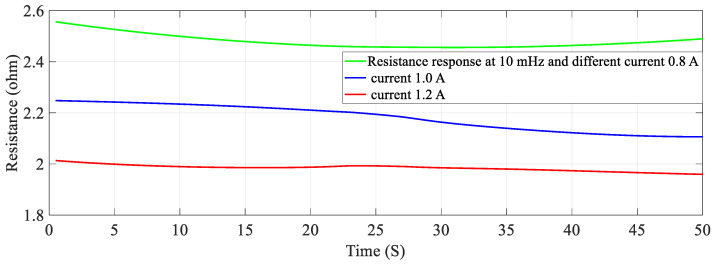
Resistance of SMA spring actuator at different current.

**Figure 8 sensors-23-02442-f008:**
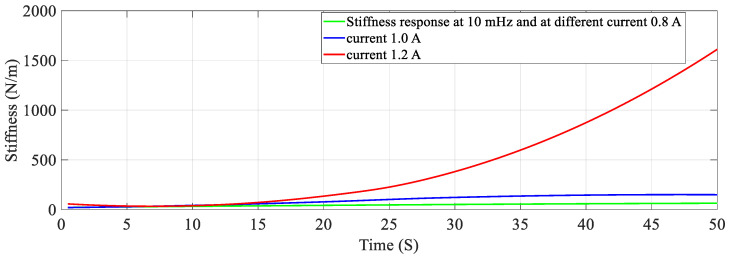
Stiffness of the SMA spring actuator at different currents.

**Figure 9 sensors-23-02442-f009:**
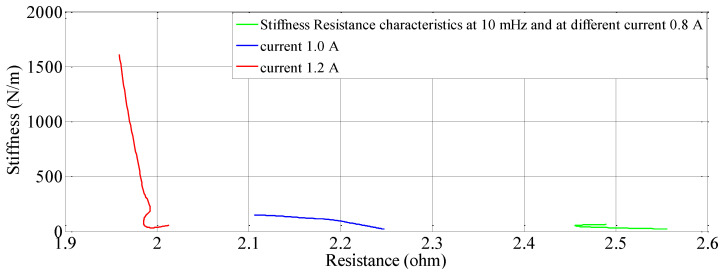
Stiffness − Resistance characteristics of the SMA spring actuator at different currents.

**Figure 10 sensors-23-02442-f010:**
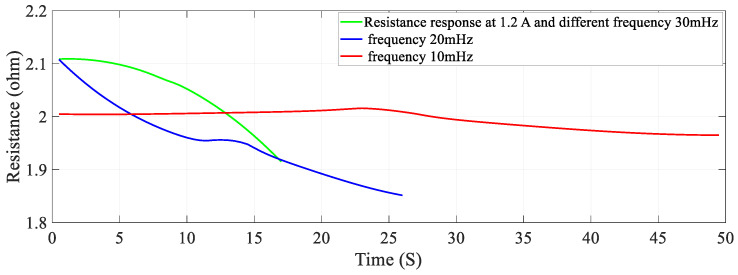
Effect of frequency on Resistance of the Shape memory Spring.

**Figure 11 sensors-23-02442-f011:**
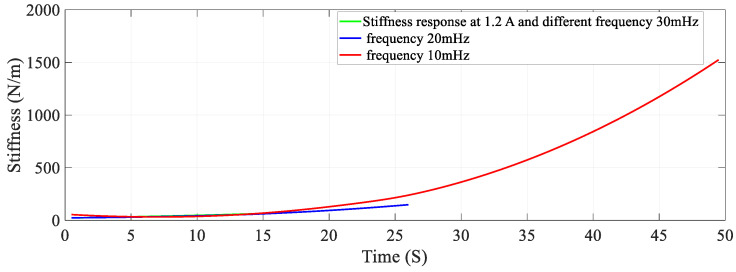
Effect of frequency on Stiffness of the Shape memory spring.

**Figure 12 sensors-23-02442-f012:**
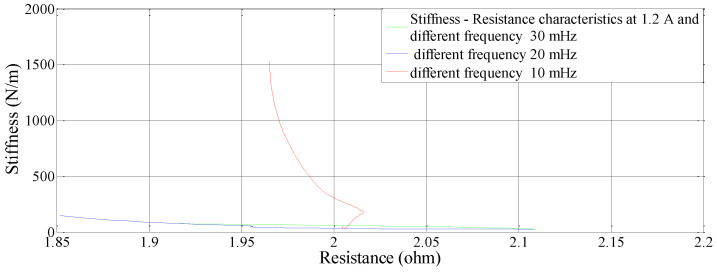
Effect of different frequency on Stiffness − Resistance Characteristics of the Shape Memory spring actuator.

**Figure 13 sensors-23-02442-f013:**
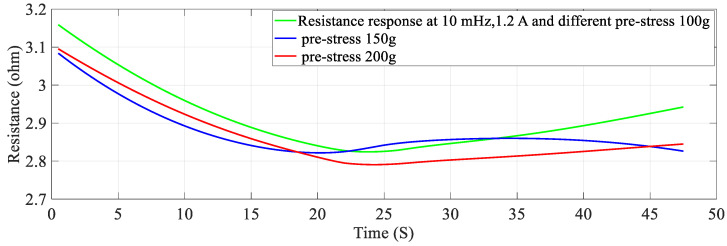
Resistance response at different Pre-Stresses.

**Figure 14 sensors-23-02442-f014:**
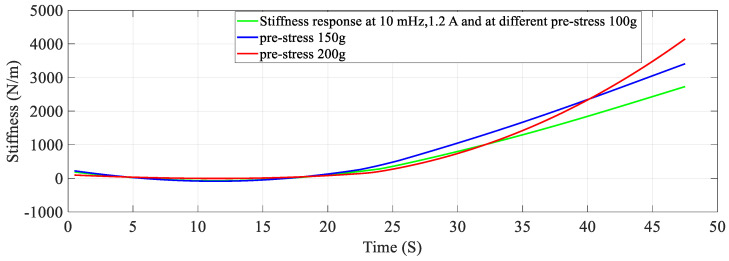
Stiffness response at different Pre−Stress.

**Figure 15 sensors-23-02442-f015:**
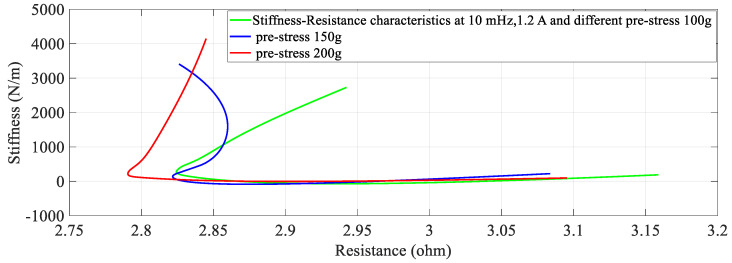
Stiffness − Resistance characteristics at different Pre−Stresses.

**Figure 16 sensors-23-02442-f016:**
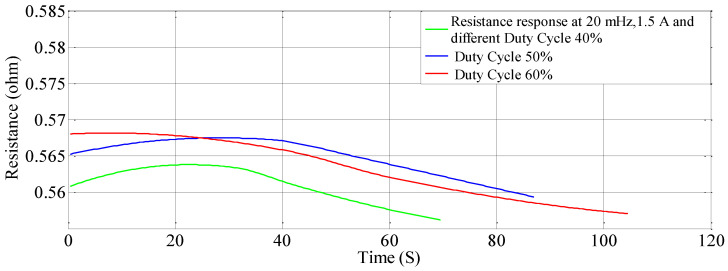
Resistance response at different duty cycles.

**Figure 17 sensors-23-02442-f017:**
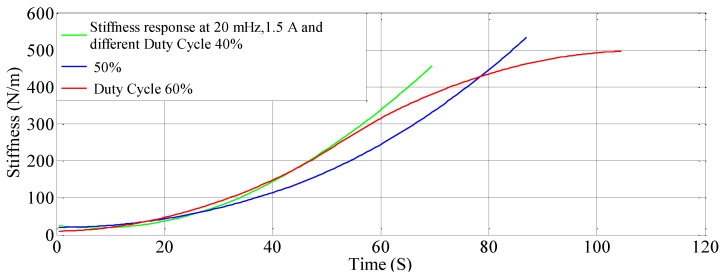
Stiffness response at different duty cycles.

**Figure 18 sensors-23-02442-f018:**
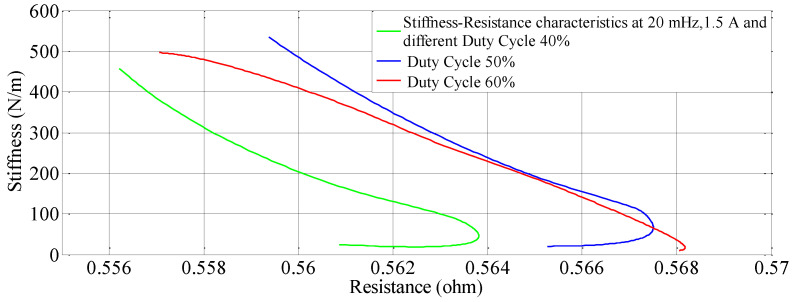
Effect of duty cycles on the Stiffness—Resistance characteristics.

**Table 1 sensors-23-02442-t001:** Technical specifications of the passive steel spring.

Physical Properties	Value
Hardness (max.)	220 HV
Yield stress (min.)	290 MN/m^2^
Tensile strength	640 MN/m^2^
Spring rate (Stiffness constant)	130 N/m

**Table 2 sensors-23-02442-t002:** Specifications of the SMC.

Physical Properties	Range/Value
Melting point	1310 °C
Electrical resistivity	76 × 10^−5^ Ω m
Modulus of elasticity	28–41 GPa
Latent heat of transformation	5.78 kCal/kg
Thermal conductivity	18.0 W/m °C
Thermal expansion coefficient	Martensite 6.6 × 10^−6^/°C Austenite 11.0 × 10^−6^/°C
Poisson ratio	0.33
Electrical resistivity	Martensite 80 × 10^−8^ Ω cm Austenite 100 × 10^−8^ Ω cm
Specific heat, C_p_	1.84 J or 0.44 kCal/kg °C
Convective heat transfer coefficient, h	54.50
Transformation temperatures:	
Ingot austenite finish (A_f_)	75 °C to 110 °C
Finished product A_f_	50 °C to 80 °C
Mechanical properties	
Ultimate tensile strength	≥1070 MPa
Total elongation	≥10%
Loading plateau stress @ 3%	≥100 MPa
Shape memory strain	≤8.0%
Geometrical parameters	
Wire diameter	5.0 × 10^−4^ m
No. of coils	45
Coil diameter	3.0 × 10^−3^ m
Volume	81.67 × 10^−8^ m^2^

**Table 3 sensors-23-02442-t003:** General properties of SMA [1].

Wire Diameter (µm)	100	125	200
Linear Resistance (Ω/m)	126	75	29
Maximum allowable force (N)	4.601	7.220	18.247
Nominal force (N)	0.275	0.422	1.079

**Table 4 sensors-23-02442-t004:** Metrics of inferential models at different currents.

	Current (A)	Type of Inferential Model	0.8 A	1.0 A	1.2A
Metrics					
MSE		Polynomial Model	8.8421 × 10^−5^	1.5728 × 10^−4^	0.0043
RMSE			0.0094	0.0125	0.0657
Goodness of Fit (R-squared) (%)			99.8991	99.8731	94.9956
MSE		Support Vector Regression	0.0025	0.0012	0.0051
RMSE			0.0500	0.0345	0.0717
Goodness of Fit (R-squared) (%)			97.1421	99.0424	94.0439

**Table 5 sensors-23-02442-t005:** Correlation Coefficient between observed and predicted stiffness at different currents.

Activation Current (A)	0.8		1.0		1.2	
Correlation Coefficient of Polynomial Model	1.0000	0.9995	1.0000	0.9994	1.0000	0.9747
	0.9995	1.0000	0.9994	1.0000	0.9747	1.0000
Correlation Coefficient of Support Vector Regression	1.0000	0.9865	1.0000	0.9961	1.0000	0.9731
	0.9865	1.0000	0.9961	1.0000	0.9731	1.0000

**Table 6 sensors-23-02442-t006:** Metrics of inferential models at different frequencies.

	Frequency (mHz)	Type of Inferential Model	10 mHz	20 mHz	30 mHz
Metrics					
MSE		Polynomial Model	0.0012	5.2667 × 10^−4^	0.0026
RMSE			0.0340	0.0229	0.0508
Goodness of Fit (R-squared) (%)			98.6401	99.3929	96.9936
MSE		Support Vector Regression	0.0025	0.0022	0.0033
RMSE			0.0496	0.0471	0.9699
Goodness of Fit (R-squared) (%)			97.1139	97.4384	96.1031

**Table 7 sensors-23-02442-t007:** Correlation Coefficient between observed and predicted stiffness at different frequencies.

Frequency (mHz)	10		20		30	
Correlation Coefficient of Polynomial Model	1.0000	0.9932	1.0000	0.9970	1.0000	0.9849
	0.9932	1.0000	0.9970	1.0000	0.9849	1.0000
Correlation Coefficient of Support Vector Regression	1.0000	0.9868	1.0000	0.9894	1.0000	0.9818
	0.9868	1.0000	0.9894	1.0000	0.9818	1.0000

**Table 8 sensors-23-02442-t008:** Metrics of Inferential models at different duty cycles.

	Duty Cycle (%)	Type of Inferential Model	40 (%)	50 (%)	60 (%)
Metrics					
MSE		Polynomial Model	0.0052	0.0066	1.5130 × 10^−4^
RMSE			0.0722	0.0811	0.0123
Goodness of Fit (R-squared) (%)			94.4167	92.4220	99.8835
MSE		Support Vector Regression	0.0053	0.0067	0.0016
RMSE			0.0731	0.0067	0.0396
Goodness of Fit (R-squared) (%)			94.2709	92.2769	98.7947

**Table 9 sensors-23-02442-t009:** Correlation coefficient at different duty cycles.

Duty Cycle (%)	40		50		60	
Correlation Coefficient of Polynomial Model	1.0000	0.9717	1.0000	0.9614	1.0000	0.9994
	0.9717	1.0000	0.9614	1.0000	0.9994	1.0000
Correlation Coefficient of Support Vector Regression	1.0000	0.9718	1.0000	0.9632	1.0000	0.9980
	0.9718	1.0000	0.9632	1.0000	0.9980	1.0000

**Table 10 sensors-23-02442-t010:** Metrics of inferential models at different pre-stresses.

	Pre-Stress (g)	Type of Inferential Model	100 g	150 g	200 g
Metrics					
MSE		Polynomial Model	0.0022	0.0018	0.0025
RMSE			0.0464	0.0421	0.0501
Goodness of Fit (R-squared) (%)			97.4866	97.9452	97.0727
MSE		Support Vector Regression	0.0100	0.0093	0.0040
RMSE			0.0998	0.0966	0.0634
Goodness of Fit (R-squared) (%)			88.3812	89.1636	95.3135

**Table 11 sensors-23-02442-t011:** Correlation coefficient at different pre-stresses.

Pre-Stress	100 g		150 g		200 g	
Correlation Coefficient of Polynomial Model	1.0000	0.9874	1.0000	0.9897	1.0000	0.9583
	0.9874	1.0000	0.9897	1.0000	0.9583	1.0000
Correlation Coefficient of Support Vector Regression	1.0000	0.9532	1.0000	0.9583	1.0000	0.9791
	0.9532	1.0000	0.9583	1.0000	0.9791	1.0000

**Table 12 sensors-23-02442-t012:** Resistance and Stiffness at different currents.

Current (A)	Resistance Change (Ω)		Stiffness Change (N/m)		Number of Cycles
	Min.	Max.	Min.	Max.	
0.8	2.4426	2.5461	21.7310	62.8684	02
1.0	2.1310	2.2530	21.7521	156.2679	02
1.2	1.9454	2.0446	23.4013	2130.4059	02

**Table 13 sensors-23-02442-t013:** Stiffness and resistance at different frequencies.

Frequency (mHz)	Resistance Change in (Ω)		Stiffness Change in (N/m)		Number of Cycles
	Min.	Max.	Min.	Max.	
10	1.94	2.04	23.25	2130.90	02
20	1.83	2.12	22.12	144.44	04
30	1.89	2.11	21.91	79.45	06

**Table 14 sensors-23-02442-t014:** Resistance and Stiffness Values at different Pre−Stresses.

Pre-Stress (g)	Resistance (Ω)		Stiffness (N/m)		Number of Cycles
	Min.	Max.	Min.	Max.	
100	2.80	3.15	24.24	2248.97	02
150	2.76	3.12	31.66	2912.84	02
200	2.70	3.05	30.79	3251.99	02

**Table 15 sensors-23-02442-t015:** Resistance and Stiffness at different Duty Cycles.

Duty Cycle (%)	Resistance (Ω)		Stiffness (N/m)		Number of Cycles
	Min.	Max.	Min.	Max.	
40	0.55	0.56	16.90	423.99	03
50	0.55	0.57	19.71	574.25	03
60	0.55	0.57	16.79	495.21	03

**Table 16 sensors-23-02442-t016:** Resistance and stiffness values at different Influencing factors.

	Properties	Shape Memory Coil Resistance Change (Ω)	Shape Memory Coil Stiffness Change (N/m)
Parameters			
Activation Current (A)		0.12	134.51
Excitation Frequency (Hz)		0.29	122.32
Pre-stress (g)		0.35	3220.20
Duty Cycle (%)		0.01	478.42

**Table 17 sensors-23-02442-t017:** Performance of SVR and nonlinear regression model prediction.

	Metrics	Average Goodness of Fit (SVM Regression)	Average Goodness of Fit (Nonlinear Regression)
Parameter			
Different Activation Currents		96.74	98.25
Different Excitation Frequencies		96.88	98.34
Different Duty Cycles		95.11	95.57
Different Pre-stress		90.95	97.50

## Data Availability

The raw/processed data required to reproduce these findings cannot be shared at this time as the data also form part of an ongoing study. In the future, however, the raw data required to reproduce these findings will be available from the corresponding authors.

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
