# Peer review of "Self-Sensing Variable Stiffness Actuation of Shape Memory Coil by an Inferential Soft Sensor"

_sensors, 2023, doi:10.3390/s23052442_

Round 1

Reviewer 1 Report

This paper reports a strategy to obtain the stiffness from the measurement of electrical resistance of a shape memory wire/fiber during variable stiffness actuation. The electrical resistance is related to the stiffness. In order to obtain the displacement, the force should be measured. Thus, the claim of a sensorless system is not rigorous.

In addition, can the authors clarify how the proposed self-sensing variable stiffness actuation method can be applicable to many control systems including robotic, biomedical, aeronautical and structural health monitoring control? It is suggested to describe some specific application scenarios. 

Some relevant publications about self-sensing functions of fibers/wires can be cited to highlight the significance of the work, such as Highly Integrated Multi‐Material Fibers for Soft Robotics; Advanced multimaterial electronic and optoelectronic fibers and textiles

Reviewer 2 Report

Ir is a very nice contribution proposing a self-sensing variable stiffness actuation method that can be applicable to many control systems. The article is written very clearly, references are suitable and the conclusions addressed are fully supported.

Reviewer 3 Report

This paper developed a support vector machine regression and nonlinear regression model to predict the stiffness of a shape memory alloy (SMA). Some experiments were conducted with different electrical and mechanical input conditions. This is an interesting topic. However, there are some issues that should be discussed further.

1. The author should describe the devices in the test bench elaborately.

2. It is difficult to understand the equations of the support vector regressor algorithm (2), (3) and (4). Authors should represent these equations according to the vector and matrix operators.

3. In equation (7), the variable, G, is calculated from force and displacement measurements using transducers. The authors should present this equation.

Reviewer 4 Report

Dear Authors,

Following are my comments:

The manuscript having ID: Sensors-2172816 which I reviewed presents a framework to obtain the stiffness of a shape memory coil during variable stiffness actuation. The contribution of the manuscript is worthy to be published. I recommend to publish this manuscript with the following modifications/responses:

 1.               Please, use the full form first and then use the abbreviation in the subsequent places in the manuscript. For instance, SMA was used in line 40 without full form and then full form was used in line 44 after.

2.               Overall the introduction does not seem in a presentable form. The paragraphs are too lengthy, and first reference has been given in line 53. I suggest to write the introduction in a very focused way. It shall cover the subject and must have a flow. Each paragraph shall not be too lengthy, and it shall impart crisp information to the readers.

3.               Follow any article for better heading. As instead of facility, use Experimental set-up etc.

4.               Figure 1 shall be improved. Labelling of the experimental set-up should be done on the figure as there is plenty of space.

5.               I believe, there is no need of figure 2. If it is really needed then enhance the quality and font size.

6.               Group figure 4, 5, and 6 in a single figure as these give same information to the reader. The only difference is current. Same is for Figures (7, 8 and 9). Figures (10,11, and 12). Figures (13,14,15), etc.

7.               Cite the following article discussing the development of self-sensing structures and sensors which is relevant to this work.

https://doi.org/10.1016/j.compstruct.2022.116169

https://doi.org/10.1016/j.coco.2022.101382

Round 2

Reviewer 4 Report

All my comments have been addressed.